mHealth technology for ecological momentary assessment in physical activity research: a systematic review

Zapata-Lamana Rafael 1
Lalanza Jaume F. 2
Losilla Josep-Maria 3 4
Parrado Eva 2 4
Capdevila Lluis 2 4 lluis.capdevila@uab.cat
1 Escuela de Educación, Universidad de Concepción , Los Ángeles , Chile
2 Department of Basic Psychology, Universitat Autónoma de Barcelona , Bellaterra , Spain
3 Department of Psychobiology and Methodology of Health Science, Universitat Autónoma de Barcelona , Bellaterra , Spain
4 Sport Research Institute UAB, Universitat Autònoma de Barcelona , Bellaterra , Spain
Morgan Amy
Electronic publication date: 2020 Mar 26
Publication date: 2020
Volume: 8
Electronic Location ID: e8848
Received 2019 Oct 16; Accepted 2020 Mar 3
Copyright: © 2020 Zapata-Lamana et al.
Copyright year: 2020
Copyright holder: Zapata-Lamana et al.
License: This is an open access article distributed under the terms of the Creative Commons Attribution License, which permits unrestricted use, distribution, reproduction and adaptation in any medium and for any purpose provided that it is properly attributed. For attribution, the original author(s), title, publication source (PeerJ) and either DOI or URL of the article must be cited.
License URL: https://creativecommons.org/licenses/by/4.0/

Keywords: mHealth, EMA, Ecological momentary assessment, Physical activity, Health, Lifestyle, Smartphones, Mobile devices, Systematic review, eHealth

Funding: Spanish Government DEP2015-68538-C2-1-R and PGC2018-100675-B-I00 Universidad de Concepción (Chile) This study was supported by the grants DEP2015-68538-C2-1-R and PGC2018-100675-B-I00 from The Spanish Government. Rafael Zapata-Lamana is supported by a scholarship from the Universidad de Concepción (Chile). The funders had no role in study design, data collection and analysis, decision to publish, or preparation of the manuscript.

==============================
Objective

To systematically review the publications on ecological momentary assessment (EMA) relating to physical activity (PA) behavior in order to classify the methodologies, and to identify the main mHealth technology-based tools and procedures that have been applied during the first 10 years since the emergence of smartphones. As a result of this review, we want to ask if there is enough evidence to propose the use of the term “mEMA” (mobile-based EMA).

Design

A systematic review according to PRISMA Statement (PROSPERO registration: CRD42018088136).

Method

Four databases (PsycINFO, CINALH, Medline and Web of Science Core Collection) were searched electronically from 2008 to February 2018.

Results

A total of 76 studies from 297 potential articles on the use of EMA and PA were included in this review. It was found that 71% of studies specifically used “EMA” for assessing PA behaviors but the rest used other terminology that also adjusted to the inclusion criteria. Just over half (51.3%) of studies (39) used mHealth technology, mainly smartphones, for collecting EMA data. The majority (79.5%) of these studies (31 out of 39) were published during the last 4 years. On the other hand, 58.8% of studies that only used paper-and-pencil were published during the first 3 years of the 10-year period analyzed. An accelerometer was the main built-in sensor used for collecting PA behavior by means of mHealth (69%). Most of the studies were carried out on young-adult samples, with only three studies in older adults. Women were included in 60% of studies, and healthy people in 82%. The studies lasted between 1 and 7 days in 57.9%, and between three and seven assessments per day were carried out in 37%. The most popular topics evaluated together with PA were psychological state and social and environmental context.

Conclusions

We have classified the EMA methodologies used for assessing PA behaviors. A total of 71% of studies used the term “EMA” and 51.3% used mHealth technology. Accelerometers have been the main built-in sensor used for collecting PA. The change of trend in the use of tools for EMA in PA coincides with the technological advances of the last decade due to the emergence of smartphones and mHealth technology. There is enough evidence to use the term mEMA when mHealth technology is being used for monitoring real-time lifestyle behaviors in natural situations. We define mEMA as the use of mobile computing and communication technologies for the EMA of health and lifestyle behaviors. It is clear that the use of mHealth is increasing, but there is still a lot to be gained from taking advantage of all the capabilities of this technology in order to apply EMA to PA behavior. Thus, mEMA methodology can help in the monitoring of healthy lifestyles under both subjective and objective perspectives. The tendency for future research should be the automatic recognition of the PA of the user without interrupting their behavior. The ecological information could be completed with voice messages, image captures or brief text selections on the touch screen made in real time, all managed through smartphone apps. This methodology could be extended when EMA combined with mHealth are used to evaluate other lifestyle behaviors.

Introduction

Physical inactivity is a leading cause of death worldwide. It is considered a pandemic in the 21st Century (Kohl et al., 2012) and is more prevalent in developed countries (Dumith et al., 2011). Unfortunately, despite the consensus on the benefits of physical activity (PA), the levels of sedentary lifestyle have increased worldwide (Booth et al., 2017). Regular PA of moderate intensity should be promoted for the entire population in order to reduce the risk of suffering many health disorders, such as cardiovascular disease, type 2 diabetes, Alzheimer’s disease or depression (Fiuza-Luces et al., 2013), and to achieve substantial health benefits (Physical Activity Guidelines Advisory Committee, 2018).

In this context, an important problem is to unify the research methodology so that scientific evidence can be recognized. On the one hand, PA is often assessed by weekly self-reports (Prince et al., 2008). The main limitation of these studies is that they are based on retrospective assessments which can lead to a recall bias (Shiffman, Stone & Hufford, 2008), and thereby errors could occur due to overestimation and underestimation of PA (Shephard, 2003). In addition, these assessments are subjective estimations by the participants, thus reliability and validity could be highly affected (Marszalek et al., 2014). On the other hand, there are objective methods available with greater reliability, such as accelerometry, GPS positioning, heart rate monitoring and movement sensors. But studies using these methods generally report PA values without examining contextual co-variables (Ainsworth et al., 2015). In fact, some studies tend to not taken in account the temporal influences (Bauman et al., 2012) nor the individual characteristics (Dunton, 2017). In order to solve these problems, the objective measures could be matched with self-reports in real time regarding the context of PA.

In this sense, Ecological Momentary Assessment (EMA) (Stone & Shiffman, 1994) is a suitable methodology which enables studies to be conducted in order to analyse lifestyle experiences in real-time, in real-world settings, over time and across contexts. EMA is based on monitoring or sampling strategies to assess phenomena at the moment they occur in natural settings. Thus, from the original proposal of this methodology, “Ecological” means that the data are captured in the natural environment of the subjects; “Momentary” means that assessments focus on current feelings and behaviors, rather than concentrating on recall or summary over long periods of time; and “Assessment” means that multiple assessments are collected over time to provide a profile for behavior throughout time (Shiffman, Stone & Hufford, 2008). A good contribution to EMA methodology has been a checklist for reporting EMA studies (CREMAS) in nutrition and PA among young people (Liao et al., 2016). However, this checklist becomes very demanding for describing published studies that, in general terms, comply with the basic features of the EMA methodology. For instance, in a recent systematic review using CREMAS to assess sedentary behavior in articles published between 2007 and 2017, only 21 of 50 potential studies were included. It is surprising that only four of these works were combining EMA with objective measurement like accelerometry (Romanzini et al., 2019), since those 10 years is when the massive appearance of smartphones took place worldwide. Since the emergence of the first mobile phone in the 1970s, and from the emergence of the first applications (apps) for iOS and Android operating systems around 2008, smartphones have rapidly evolved. The improvement in fast processors, small and long-lasting batteries, large memory capacity and very precise built-in sensors has paved the way for apps that are now affecting our lifestyle (Ozdalga, Ozdalga & Ahuja, 2012).

Thus, in the last years, we are talking about mobile health (mHealth) when we are using some of the smartphone capabilities for assessing or monitoring health or lifestyle (Fiordelli, Diviani & Schulz, 2013). More specifically, the term mHealth has been defined as “the use of mobile computing and communication technologies in health care and public health” (Free et al., 2010). For example, you can use the camera of a smartphone to capture images for assessing dietary intake (Daugherty et al., 2012); or mobile phone’s built-in motion sensors and self-report through the touch screen for measuring PA (Dunton et al., 2014a).

Existing systematic reviews of EMA interventions focus on specific age groups such as young people (Marszalek et al., 2014; Liao et al., 2016); or about aspects of the lifestyle other than the PA, such as emphasizing sedentary behavior (Romanzini et al., 2019); or are based on an excessively strict list of criteria that does not include all EMA studies according to their original definition (Liao et al., 2016; Romanzini et al., 2019).

Our objective is to systematically review the scientific publications on EMA relating to PA behavior in order to classify the methodologies and to identify the main mHealth technology-based tools and procedures that have been applied during the first 10 years since the emergence of smartphones. As a result of this review we want to ask if there is enough evidence to propose the use of the term “mEMA” (mobile-based EMA) when mHealth technology is being used together with the EMA methodology for monitoring lifestyle behaviors such as PA, in real time and in natural environments.

Methodology

The study was undertaken in accordance with the Preferred Reporting Items for Systematic Reviews and Meta-Analyses (PRISMA) statement (Liberati et al., 2009). A systematic review protocol had previously been registered in the PROSPERO repository with the code: CRD42018088136 (Zapata-Lamana et al., 2018).

Search strategy

A systematic review of the literature was performed using the following databases and portals according to the order indicated: PsycINFO by PsycNET, CINAHL by EBSCOhost, MEDLINE by PubMed, and Core Collection of Web of Science by Web of Science. Our aim was to identify studies that used EMA methodologies to measure PA in participants of all ages. The search comprised the period between 2008 and 2018 because we were interested in analyzing the use of mHealth technology to evaluate PA behavior. This is because mHealth solutions are based on applications for smartphones (apps) and we have already commented that the first apps appeared around 2008. According to Liao et al. (2016), an increasing number of studies on PA have adopted EMA due to methodological advances in mobile technologies in recent years. Thus, we consider that before 2008 smartphones or tablets still did not have enough capable built-in sensors for allowing EMA.

When possible, keywords and terms were obtained from Thesaurus. The search strategy followed the guidelines from Peer Review of Electronic Search Strategies (PRESS) (McGowan et al., 2016). The searching general syntax was: (“EMA” OR “experience sampling” OR “experience samplings”) AND (Exercise OR Exercises OR “PA” OR “physical activities”), and it was adapted to each database (the specific search syntaxes are provided in a Supplemental File).

Study selection and inclusion criteria

No exclusion criteria were applied for gender, age or clinical condition, but as regards the language, only full-text articles in English or Spanish were reviewed. Reviews, editorials, protocols and theses were not included. The articles selected by title and abstract met the conditions indicated in Table 1. EMA could be applied to PA or to other variables, but then PA has to be assessed by other methodologies and compared to the main EMA variable.

Table 1 Inclusion criteria and description for this review.

Criteria	Description	
(1) EMA has to accomplish the following points	(a) Instruments that collect data in real time. Participants reported the activities and/or, moods, etc. at the moment they are experiencing them or up to 24 h, after the activity was carried out	
(b) In a natural environment	
(c) Repeated measures (2 or more measures)	
(d) Self-reports and/or automatic recordings	
(e) Using both electronic devices and/or paper and pencil format	
(2) PA is considered	PA has to be spontaneous or planned activity, carried out individually or collectively and that incorporates a physical effort component of any intensity	
(3) Type of article	All articles that provide original data on the use of EMA in PA (criteria 1 and 2) and are published in a scientific journal without taking into account the type and number of sample as well as the experimental design	
Note:

EMA, Ecological Momentary Assessment; PA, Physical Activity.

Data extraction

In a first step, duplicate articles from the four databases were deleted using Mendeley. One reviewer (Rafael Zapata-Lamana) applied the inclusion/exclusion criteria to all titles and abstracts. Articles meeting the inclusion criteria were selected and when decisions could not be made from the title and abstract alone, the full article was also retrieved. The selected papers were checked independently by two review authors (Rafael Zapata-Lamana, Jaume F. Lalanza). Discrepancies were resolved through discussion (with a third author where necessary, Lluis Capdevila) until reaching consensus. A standardized, pre-piloted form was used to extract data from the included articles in order to assess the study quality and for the synthesis of the evidence. Extracted information included: general information (author, year, country); sample (size, population, age, etc.); details of tools used (measure, purpose/use, type of tool, etc.); methodological protocol (experimental design, response rate, time interval required to completion, and mode of administration, sensor use, etc.) as well as assessing the main variables. Furthermore, the data extraction was carried out by two reviewers (Rafael Zapata-Lamana, Jaume F. Lalanza).

Risk of bias assessment tools

The tool proposed by National Institute for Health and Care Excellence (NICE) for prognostic studies (National Institute for Health and Care Excellence, 2012) was applied by two reviews (Rafael Zapata-Lamana, Jaume F. Lalanza) to assess the risk of bias (RoB) of the selected non-experimental studies. Following the same procedure, to assess the RoB of the selected experimental and quasi-experimental studies we applied the “Cochrane Risk of Bias Tool for Randomized Trials” (Higgins et al., 2011) and the “Risk Of Bias In Non-randomized Studies—of Interventions” (ROBINS-I) (Sterne et al., 2016), respectively. Since the aim of the systematic review was not to analyze the obtained results, the following items were removed: from NICE 1.3 (“the prognostic factor of interest is adequately measured in study participants, sufficient to limit potential bias”) and 1.6 (“the statistical analysis is appropriately accounted for, limiting potential bias with respect to the prognostic factor of interest”); from Cochrane “Selective Reporting” and “Other Bias”; and from ROBINS-I “Bias in measurement of outcomes” and “Bias in selection of the reported result”.

Strategy for data synthesis

We provide a narrative synthesis of the findings from the included studies structuring it around the methods and the procedure related to EMA in PA research. The main information is also showed in the Tables. In the Discussion, an analysis is made on the combination of EMA and mHealth technology and some suggestions are given for future researchers in order to standardize the application of this mobile-based methodology for monitoring PA behaviors.

Results

Literature search

Figure 1 shows the flow diagram of systematic reviews for analysis of studies proposed by PRISMA. After the duplicate records in databases were excluded, a total of 297 potential studies as regards EMA in PA were identified. The eligibility criteria were applied and the analysis was made by reading the title and the abstracts. During the selection phase, all discrepancies between the two reviewers were resolved by consensus without requiring a third reviewer. Finally, 74 articles were considered for this review for the qualitative synthesis of the data. In fact, a total of 76 studies were analyzed because two of those 74 articles contained 2 studies with each one with different samples.

Figure 1 PRISMA flow diagram with the different previous phases to the qualitative synthesis.

Studies characteristics with EMA in PA

Table 1 described the inclusion criteria for this review. In our systematic review, we identified 54 of 76 studies (71%) that specifically used “EMA” for assessing PA behaviors. But there were 29% studies that used other terminology that also adjusted to the inclusion criteria. Thus, we identified studies that used “Experienced Sampling Method” (ESM) (Bohnert et al., 2013; Cabrita et al., 2017; Fuller-Tyszkiewicz, Skouteris & Mccabe, 2013; Heininga et al., 2017; McCormick et al., 2008, 2009; Snippe et al., 2016; Wichers et al., 2015; Salvy et al., 2008) for referring to a similar methodology. We also found other terms used in a similar way, like “Ambulatory Assessment” was used in 6 studies (Bossmann et al., 2013; Kanning, Ebner-Priemer & Schlicht, 2015; Kanning & Schoebi, 2016; Kanning & Hansen, 2017; Timmerman et al., 2015; Von Haaren et al., 2013), with “Naturalistic Study”, only used in 1 study (Bonham, Pepper & Nettle, 2018). Finally, we found 5 studies that applied a similar methodology without specifically naming it (Aggio et al., 2017; Hyde et al., 2011; LePage & Crowther, 2010; Mata et al., 2012; Sternfeld et al., 2012).

The publication date of articles ranged between 2008 and 2018. With 19.7% of articles being published between 2008 and 2011, 39.5% between 2012 and 2015, and 38.2% between 2016 and 2018. When clustering the studies geographically, 63.2% of these were performed in North America (including the USA), 34.2% in Europe, 5.3% in Asia and 1.3% in Oceania, with the USA being the country with the highest publication rate (56.6%). The majority of studies included both males and females, with a total number of 13.663 participants (60.2% females).

As regards the characteristics of the participants (Table 2), 25% of studies were carried out with children and adolescents (9–17 years), with a total number of 8,543 participants (55.7% females). On the one hand, 26.3% of studies were carried out with university students, with a total number of 1388 participants (67.4% females), and an age range between 18 and 28 years. On the other hand, 44.7% of studies were carried out with 3,585 adult participants (69.8% females), an age range between 18 and 59 years, with heterogeneous characteristics (patients, healthy subjects, active, inactive, recreational runners, etc.). Only 3 studies (3.9%) were carried out with 147 older adult participants (49.7% females), with an age range between 60 and 73 years. As regards the designs of the studies, 92.1% used a longitudinal prospective design. Only 5 studies (6.6%) used a randomized-experimental design and only 1 study used a quasi-experimental design.

Table 2 General characteristics of reviewed studies about ecological momentary assessment (EMA) in physical activity (PA).

Reference, year, country	Study population	n (males)	Age	Design	
Aggio et al. (2017), England	US	51 (26)	24.0 ± 4.7	L, P, NI	
Basen-Engquist et al. (2013), USA	Women Endometrial Cancer Survivor*	79 (0)	57.0 ± 11.01	L, P, NI	
Biddle et al. (2009c), Hun, Slo, Ro	Adolescents	623 (247)	15.5 ± 0.9	L, P, NI	
Biddle, Gorely & Marshall (2009), UK	Adolescents	1,484 (561)	14.67 ± 0.92	L, P, NI	
Biddle et al. (2009b), UK	Adolescents	1,493 (566)	14.8	L, P, NI	
Biddle et al. (2009a), Scotland	Adolescents	991 (385)	14.1	L, P, NI	
Bohnert et al. (2013), USA	Adolescents	25 (9)	12.6 ± 1.0	L, P, NI	
Bond et al. (2013), USA	Bariatric Surgery Patients*	21 (4)	48.5 ± 2.8	L, P, NI	
Bonham, Pepper & Nettle (2018), England	Adults (Recreational Runners)	38 (20)	18–49	L, P, NI	
Bond et al. (2013), Germany	US	62 (53)	21.4 ± 1.8	L, P, NI	
Brannon et al. (2016), USA	Adolescents	20 (12)	15.67 ± 1.75	L, P, NI	
Burg et al. (2017), USA	Adults (eager to became active)	63 (27)	31.9 ± 9.5	L, E, CG, R	
Brannon et al. (2016), Netherlands	Older adults	10 (4)	68.7 ± 5.5	L, P, NI	
Conroy et al. (2013), USA	US	128 (53)	21.3 ± 1.1	L, P, NI	
Cabrita et al. (2017), USA	Children	121 (62)	9–13	L, P, NI	
Conroy et al. (2013), USA	Children	108 (59)	11	L, P, NI	
Dunton et al. (2011), USA	Children	94 (49)	9–13	L, QE, CG	
Dunton et al. (2012a), USA	Adults	110 (30)	40.4 ± 9.74	L, P, NI	
Dunton et al. (2012b), USA	Children	114 (56)	9–13	L, P, NI	
Dunton et al. (2012c), USA	Adults	116 (32)	40.5 ± 9.5	L, P, NI	
Dunton et al. (2014b), USA	Adolescents	39 (18)	15.9 ± 1.2	L, P, NI	
Elavsky, Kishida & Mogle (2016), USA	Women (Peri-post menopausal)	121 (0)	51.5	L, P, NI	
Emerson, Dunsiger & Williams (2018), USA	Adults (Low-Active and Obese)*	59 (7)	47.71 ± 11.06	L, E, CG, R	
Fanning et al. (2016), USA	US	33 (9)	20.5 ± 1.5	L, P, NI	
Fortier et al. (2015), Canada	Women (Physically active Mothers)	63 (0)	42.6 ± 5.61	L, P, NI	
Fuller-Tyszkiewicz, Skouteris & Mccabe (2013), Australia	Adult Women	37 (0)	34.05 ± 9.73	L, P, NI	
Gorely et al. (2009a), UK	Adolescents	1,171 (477)	14.8 ± 0.86	L, P, NI	
Gorely et al. (2009b), UK	Adolescent boys	561 (561)	14.6 ± 0.89	L, P, NI	
Hager et al. (2017), USA	Mother-Toddler Dyads	160 (0)	26.6	L, P, NI	
Hausenblas et al. (2008)	US	40 (14)	20.5 ± 2.5	L, E, R	
Heininga et al. (2017), Netherlands	Adults (with or without anhedonia)*	138 (28)	21.48	L, P, NI	
Hekler et al. (2012), USA	Adults	14 (7)	59.4 ± 6.4	L, P, NI	
Hyde et al. (2011), USA	US	190 (65)	19.3 ± 2.8	L, P, NI	
Jones et al. (2017), USA	Adults (stressed)*	105 (29)	40.3 ± 9.8	L, P, NI	
Kanning, Ebner-Priemer & Schlicht (2015), Germany	Older adults	69 (35)	60.1 ± 7.1	L, P, NI	
Kanning & Schoebi (2016), Germany	US	65 (28)	24.6 ± 3.2	L, P, NI	
Kanning & Hansen (2017), Germany	Older adults	68 (35)	60.1 ± 7.1	L, P, NI	
Kim et al. (2014), Japan	Depressive patients and healthy adults*	57 (55)	37.35	L, P, NI, CG	
Knell et al. (2017), USA	Adults	238 (78)	43.3	L, P, NI	
LePage et al. (2012), USA	–	–	–	–	
Study 1	US	51 (0)	19.06 ± 3.10	L, P, NI	
Study 2	US with eating disorders*	76 (0)	19.08 ± 2.86	L, P, NI, CG	
LePage & Crowther (2010), USA	US (with body dissatisfaction)	54 (0)	19.1 ± 2.88	L, P, NI, CG	
Liao et al. (2014), USA	Children and Adolescents	120 (62)	9–13	L, P, NI	
Liao, Intille & Dunton (2015), USA	Adults	114 (30)	27–73	L, P, NI	
Liao et al. (2017a), USA	Adults	82 (22)	39.8	L, P, NI	
Liao et al. (2017b), USA	Adults (dog owners)	71 (17)	40.2 ± 8.6	L, P, NI	
Liao, Solomon & Dunton (2017), USA	Adults	110 (30)	40.4 ± 9.74	L, P, NI	
Maher et al. (2014), USA	US	128 (53)	21.3 ± 1.1	L, P, NI	
Maher et al. (2016), USA	Adults	90 (30)	40.3 ± 9.6	L, P, NI	
Marquet et al. (2017), USA	US (players of Pokemon Go)	47 (24)	19.45	L, P, NI	
Marquet, Alberico & Hipp (2018), USA	US (players of Pokemon Go)	74 (37)	19.6	L, P, NI	
Mata et al. (2012), USA	Adults with depression and controls*	106 (32)	26.8	L, P, NI, CG	
Marquet et al. (2017), USA and Ser	Adults with severe mental illness*	22 (12)	38.95	L, P, NI	
Marquet, Alberico & Hipp (2018), USA and Ser	Adults with severe mental illness*	22 (Not inf)	Not inf	L, P, NI	
Nadell et al. (2015), USA	Adults (Tabacoo smokers)	188 (88)	21.32 ± 0.77	L, P, NI	
O’Connor et al. (2017), USA	Mother Children Dyads	175 (84)	41.1	L, P, NI	
Pickering et al. (2016), USA	Adults	103 (30)	40.3 ± 9.6	L, P, NI	
Nadell et al. (2015), England	US	84 (46)	19.85	L, P, NI	
Rusby et al. (2014), USA	Adolescents	82 (40)	Not inf	L, P, NI	
Sala et al. (2017), USA	US with eating disorders*	129 (0)	19.19 ± 1.40	L, P, NI	
Salvy et al. (2008), USA	Adolescents (overweight)	20 (10)	13.52	L, P, NI, CG	
Schöndube, Kanning & Fuchs (2016), Germany	US	60 (20)	23.5 ± 2.8	L, P, NI	
Schöndube et al. (2017), Germany	US	63 (21)	23.5 ± 2.8	L, P, NI	
Seto et al. (2014), China	US and their friends	12 (Not inf)	18–31	L, P, NI	
Seto et al. (2016), China	US	12 (4)	24.6 ± 3.06	L, P, NI	
Snippe et al. (2016), Netherlands	Depressive patients*	85 (47)	48 ± 10.2	L, E, CG, R	
Soos et al. (2012), Ro and Slo	Adolescents	635 (251)	16 ± 1.0	L, P, NI	
Soos et al. (2014), UK, Hu, Ro, Slo, Chi	Adolescents	812 (348)	15.6 ± 1.0	L, P, NI	
Spook et al. (2013), Netherlands	Adolescents and Adults	30 (13)	16–21	L, P, NI	
Sternfeld et al. (2012), USA	–	–	–	–	
Pilot	Adults	33 (23)	55.6 ± 8.8	L, P, NI	
Validation	Adults	345 (296)	56.9 ± 5.7	L, P, NI	
Thomas et al. (2011), USA	Gastric patients*	21 (4)	48.5 ± 12.6	L, P, NI	
Timmerman et al. (2015), Netherlands	Cancer Survivors*	36 (12)	55.95	L, P, NI, CG	
Von Haaren et al. (2013), Germany	US	29 (Not inf)	21.3 ± 1.7	L, P, NI	
Wichers et al. (2015), Belgium	Women (Twins)	504 (0)	27 ± 7.6	L, P, NI	
Williams et al. (2016), USA	Sedentary and obese adults	59 (7)	47.7 ± 11.1	L, E, CG, R	
Notes:

* With physical diseases or psychological disorders.

Hu, Hungary; UK, United Kingdom; Slo, Slovenia; Ro, Romania; Ser, Serbia, Chi, China; US, University students; L, Longitudinal; P, Prognostic; NI, No intervention; CG, Control group; QE, Quasi Experimental; R, Randomization.

Characteristics of the main methodological aspects about EMA

The detailed characteristics of the main methodological aspects about EMA in PA are presented in Table 3. As regards the study duration, 57.9% of studies were performed between 1 and 7 days, 25% of studies were performed for a duration of 1–2 weeks. Six studies (7.9%) were performed for a duration of 3–4 weeks, and another 7 studies (9.2%) took longer than 1 month. In relation to the daily period, 46.1% of studies performed measurements all day (46.1%) and 10.5% of studies took measurement for only half a day. An individualized time of collection was used in 32.9% of studies, taking into account the timetable of each participant (e.g., studies with children sample avoided the school time). Only 10.5% of studies did not report any daily measurement. As regards the frequency of the measurements during a day, 36.8% of studies used 3 to 7 prompts per day, 22.4 of studies used 8 to 12 prompts per day, 21.1% used more than 13 prompts per day, 15.8% used 1 to 2 prompt per day. Only in 3 studies (3.9%) participants were asked to complete the EMA immediately after a target event (based on event).

Table 3 Characteristics of the main methodological aspects of EMA.

Reference	Study duration (d)	Day period	Prompts per day	EMA named	EMA instruments	EMA sensors	Topics evaluated in addition to PA	
Aggio et al. (2017)	7	Each evening	Each 24 h	No	OSS	Acc	PS, OV	
Basen-Engquist et al. (2013)	5	Morning and evening	Each 12 h	Yes	PDA	Acc	BMI, BPA, OV	
Biddle et al. (2009c)	4	All day	15 min	Yes	P&P	No	GA, ISD, OV	
Biddle, Gorely & Marshall (2009)	4	All day	15 min	Yes	P&P	No	GA	
Biddle et al. (2009b)	4	All day	15 min	Yes	P&P	No	EB, SLP	
Biddle et al. (2009a)	4	All day	15 min	Yes	P&P	No	GA	
Bohnert et al. (2013)	14	During all day	Ran (8 × d)	No, ESM	P&P	Acc	BMI, EB, GA, SC	
Bond et al. (2013)	6	At evening	Each 24 h	Yes	PDA	No	BPA, EB, ISD, OV	
Bonham, Pepper & Nettle (2018)	42	During all day	Ran (1 a × d)	No, NS	mHealth	Gps	PS, OV	
Bossmann et al. (2013)	7	During all day	1 h	No, AA	mHealth	Acc	PS	
Brannon et al. (2016)	20	Morning and evening	Depending on participants	Yes	mHealth	Acc, HR CH-B	EB, ISD, PS, SLP, SC, OV	
Burg et al. (2017)	365	During all day	3 a × d	Yes	mHealth	Acc	ISD, PS, SC	
Cabrita et al. (2017)	38	All day	1 h	No, ES	mHealth	Acc	BMI, GA, EC, GA, SC	
Conroy et al. (2013)	14	Evening	24 h	Yes	OSS	Acc	EB, GA, SLP, SC	
Dunton et al. (2011)	4	Afternoon and evening (Fri/Mon)	Ran (20 a × d)	Yes	mHealth	Acc	BMI, EC, GA, ISD, PS, SC, OV	
Dunton et al. (2012a)	8	Afternoon and evening (Fri/Mon)	Ran (20 a × d)	Yes	mHealth	Acc	BMI, EC, GA, ISD, OV	
Dunton et al. (2012b)	8	Afternoon and evening (Fri/Mon)	Ran (20 a × d)	Yes	mHealth	Acc	BMI, EC, GA, ISD, SC, OV	
Dunton et al. (2012c)	4	All day (Sat/Tu)	Ran (8 a × d)	Yes	mHealth	Acc	BMI, GA, ISD, OV	
Dunton et al. (2014a)	8	Afternoon and evening (Fri/Mon)	Ran (20 a × d)	Yes	mHealth	Acc	BMI, GA, ISD, PS	
Dunton et al. (2015)	12	All day (Sat/Tu)	Ran (8 a × d)	Yes	mHealth	Acc	BMI, EC, ISD, GA, PS, SC, OV	
Dunton, Dzubur & Intille (2016)	14	All day	Ran (3 or 7 a × d)	No, CS-EMA	mHealth	Acc	BMI, GA, SC, OV	
Elavsky, Kishida & Mogle (2016)	15	All day	Ran (4 a × d)	Yes	PDA	Acc	BMI, GA, ISD, PS	
Emerson, Dunsiger & Williams (2018)	62	Individual Schedule	24 h	Yes	PDA	No	BMI, ISD, PS	
Fanning et al. (2016)	7	All day	16 hourly prompts (1 h)	Yes	mHealth	Acc	ISD, LEP, PS	
Fortier et al. (2015)	14	At evening	24 h	No, DDM/ES	P&P	No	ISD, PS, SLP	
Fuller-Tyszkiewicz, Skouteris & Mccabe (2013)	7	All day	Ran (6 a × d)	No, ESM	PDA	No	BMI, OV	
Fortier et al. (2015)	4	All day	15 min	Yes	P&P	No	EC, GA, ISD, SC	
Gorely et al. (2009b)	4	All day	15 min	Yes	P&P	No	EC, GA, ISD, SC	
Hager et al. (2017)	8	All day	Ran (max 8 a × d)	Yes	PDA	Acc	EC, ISD, GA, OV, SC, SLP,	
Hausenblas et al. (2008)	6	During all day	Ran (3 a × d)	Yes	P&P	No	PS	
Heininga et al. (2017)	30	During all day	Ran (3 a × d)	No, ESM	mHealth	No	ISD, SC	
Hekler et al. (2012)	56	Afternoon and evening	2 a × d	Yes	PDA	No	BPA, EC, ISD, PS, OV, SC	
Hyde et al. (2011)	7	Evening	24 h	No	P&P	No	GA, PS, SLP	
Jones et al. (2017)	4	Not Informed	Ran (8 a × d)	Yes	mHealth	Acc	BMI, GA, ISD, OV, PS	
Kanning, Ebner-Priemer & Schlicht (2015)	3	Not Informed	Accelerometer was active	No, AA	mHealth	Acc	ISD, PS	
Kanning & Schoebi (2016)	1	All day	45 min	No, AA	PDA	Acc	PS	
Kanning & Hansen (2017)	3	All day	40–100 min	No, AA	mHealth	Acc	BMI, ISD, OV, PS, SC	
Kim et al. (2014)	37	During all day	Semi Ran (4 × d)	Yes	PDA	Acc	PS	
Knell et al. (2017)	7	Morning	24 h	Yes	mHealth	Acc	BMI, GA, ISD	
LePage et al. (2012)	–	–	–	–	–	–	–	
Study 1	10	During all day	Ran (4 a × d)	Yes	PDA	No	BMI, EB, OV, PS	
Study 2	7	Evening	24 h	Yes	PDA	No	EB, OV, PS	
LePage & Crowther (2010)	10	During all day	Ran (4 a × d)	Yes	PDA	No	BMI, PS, OV	
Liao et al. (2014)	4	Morning and afternoon	Ran (3 week, 7 weekend)	Yes	mHealth	No	BMI, GA, ISD, SC	
Liao, Intille & Dunton (2015)	4	Pre-programed intervals	Ran (8 a × d)	Yes	mHealth	Acc	BMI, EC, GA, ISD, OV, SC	
Liao et al. (2017a)	4	Not informed	Ran (8 a × d)	Yes	mHealth	Acc	BMI, ISD, OV, PS	
Liao et al. (2014)	12	All day	Ran (8 a × d)	Yes	mHealth	No	ISD, OV, PS	
Liao, Intille & Dunton (2015)	4	All day	Ran (8 a × d)	Yes	mHealth	Acc	BMI, ISD, OV, PS	
Maher et al. (2014)	14	Evening	24 h	Yes	P&P	Acc	BMI, OV	
Maher et al. (2016)	12	All day	Ran (8 a × d)	Yes	mHealth	Acc	BMI, BPA, ISD, OV	
Marquet et al. (2017)	7	During all day	3 a × d	Yes	mHealth	Acc	EC, ISD, OV, SC	
Marquet, Alberico & Hipp (2018)	7	During all day	3 a × d	Yes	mHealth	Acc	EC, ISD, OV, SC	
Mata et al. (2012)	7	All day	Ran (8 a × d)	No	PDA	No	OV, PS	
Marquet et al. (2017)	7	All day	Ran (7 a × d)	No, ESM	P&P	Acc	EC, GA, ISD, PS, SC	
Marquet, Alberico & Hipp (2018)	7	Not informed	Ran (7 a × d)	No, ESM	P&P	Acc	EC, SC	
Nadell et al. (2015)	7	Not informed	Ran (5–7 a × d)	Yes	PDA	No	BMI, ISD, OV PS	
O’Connor et al. (2017)	8	During all day	Ran (week 3/4, weekend 7/8)	Yes	mHealth	No	BMI, EB, ISD, OV	
Pickering et al. (2016)	4	All day	Ran (8 a × d)	Yes	mHealth	Acc	BMI, BPA, EC, ISD, OV	
Rouse & Biddle (2010)	2	Not informed	15 min	Yes	P&P	No	GA, ISD	
Rusby et al. (2014)	28	During all day	Ran (3–6 a × d)	Yes	mHealth	No	GA, ISD, PS, SC	
Sala et al. (2017)	7	Personalized for participants	Ran (4 a × d)	Yes	mHealth	No	EB, OV, PS	
Salvy et al. (2008)	7	All day	2 h	No, ESM	PDA	No	BMI, SC	
Schöndube, Kanning & Fuchs (2016)	20	During all day	Ran (4 a × d)	Yes	mHealth	No	ISD, PS	
Schöndube et al. (2017)	20	During all day	Ran (4 a × d)	Yes	mHealth	No	ISD, OV	
Seto et al. (2014)	7	Not informed	Each meal	Yes	mHealth	GPS, Acc	EB, EC, PS	
Seto et al. (2016)	14	Not informed	Ran (5 a × d)	Yes	mHealth	GPS, Acc	BMI, EB, EC, ISD, PS, OV, SC, SLP	
Snippe et al. (2016)	12	All day	Semi Ran (10 a × d)	No, ESM	mHealth	No	GA, ISD, PS, SC	
Soos et al. (2012)	4	All day	15 min (the end of day)	Yes	P&P	No	EC, GA, ISD, SC	
Soos et al. (2014)	3	All day	15 min (the end of day)	Yes	P&P	No	EC, GA, ISD, SC	
Spook et al. (2013)	7	During all day	Ran (5 a × d)	Yes	mHealth	No	BPA, EC, GA, OV, PS, SC	
Sternfeld et al. (2012)	–	–	–	–	–	–	–	
Pilot	12	During all day	3 a × d	No	mHealth, P&P	No	BMI, GA, ISD	
Validation	14	During all day	3 a × d	No	mHealth	Acc	BMI, GA, ISD	
Thomas et al. (2011)	6	All day	Semi Ran (6 a × d)	Yes	PDA	No	BMI, EB, GA, ISD,	
Timmerman et al. (2015)	5	During all day	3 (13:00, 17:00, 20:00)	No, AA	mHealth	Acc	BMI, ISD, OV	
Von Haaren et al. (2013)	2	During all day	2 h a little bit Ran	No, AA	PDA	Acc	BMI, PS	
Wichers et al. (2015)	5	All day	Semi Ran (10 a × d)	No, ESM	P&P	No	GA, ISD, PS, OV	
Williams et al. (2016)	42	All day	12 × d	Yes	PDA	No	BMI, ISD, PS	
Note:

Study duration column: d: days. Day period column: Sat: Saturday, Tu: Tuesday, Mo: Monday, Fri: Friday. Prompts per day column: h: hours, m: minutes, a × d: assessment per day, Ran: Random. EMA named column: ESM: Experienced Sampling Method, NS: Naturalistic Study, AA: Ambulatory Assessment, ES: Experience Sampling, CS-EMA: Context-Sensitive EMA, DDM/ES: Daily Diary Method or Experience Sampling. EMA instruments and sensors columns: mHealth (includes: Smartphone; iPod, Video Recording; Automatized call survey; SMS), PDA: Personal Digital Assistant or similar apparatus like watch-type computer, OSS: Online Survey System, P&P, Paper and Pencil; ACC, Accelerometers; GPS, Global Positioning System; HRCH-B, Heart rate Chest Band. Topic column, BMI, Body Mass Index, ISD: Income and Socio Demographics; GA, General Activity (included: sedentary behavior, time with the smartphone, time reading, eating, active traveling), PS: Psychological State, BPA: Barriers to Physical Activity, SLP: Sleep, EB: Eating Behavior, SC: Social Context, EC: Environmental Context, OV: Other Variables (included: self-efficacy, type of country, arousal, acceptability, safety, traffic, use of smartphone during exercise, toddler with mother, waist circumference, autonomic, competencies, vegetation, cultural variables and body perceptions, amount of garbage, energy and fatigue).

According to column “Use the name EMA” in Table 3, 71% of studies used the EMA expression for naming the main methodology, with 13.2% of studies using “Experienced Sampling Method” (ESM), and 7,9% used “Ambulatory Assessment”. Only 1 study used the expression “Naturalistic Study”, and the rest (6.6%) did not use any specific nomenclature to name the methodology of the study.

As regards the recording system for data collection, 39 studies (51.3%) used mHealth technology (mainly smartphones), 18 studies (23.7%) used PDAs (personal digital assistant, also known as a handheld PC), 18 studies (23.7%) used the most traditional system of paper and pencil, and only 2 studies (2.6%) used an online survey system. A large majority (79.5%) of the studies (31 of 39) were published during the last 4 years of the 10-year period analyzed between 2014 and February 2018. On the other hand, 76.5% of the studies that only used paper and pencil to collect EMA data were published before 2014, and 58.8% during the three 1st years of the 10-year period analyzed between 2008 and February 2010 (Biddle, Gorely & Marshall, 2009; Biddle et al., 2009a, 2009b, 2009c; Gorely et al., 2009a, 2009b; Hausenblas et al., 2008; McCormick et al., 2008, 2009; Rouse & Biddle, 2010). For its part, 11 of 18 studies (61.1%) that only used PDAs were published during the middle years of the 10-year period analyzed between 2011 and 2015 (Basen-Engquist et al., 2013; Bond et al., 2013; Fuller-Tyszkiewicz, Skouteris & Mccabe, 2013; Hekler et al., 2012; Kim et al., 2014; Mata et al., 2012; Nadell et al., 2015; Thomas et al., 2011; Von Haaren et al., 2013; LePage & Crowther, 2010).

As regards the sensors for collecting data (see column “EMA sensors” in Table 3), the accelerometer was used in 39 studies (51.3%), and 27 of these studies used the accelerometer built-in to the smartphone as an own sensor (mHealth technology). Two of them used the accelerometer combined with GPS using mHealth technology. Only 1 study used GPS only (with mHealth), and the rest of studies did not use any sensor to collect data. 81.5% of studies using mHealth with accelerometers were published between 2014 and 2018. The rest of the studies that recorded PA with accelerometry but not with mHealth technology (12 studies) used conventional accelerometers for the study combined with PDA, paper and pencil, and online survey recordings. Only 1 study used a cardiac chest band for recording heart rate with mHealth. Finally, it was found 25 studies did not use mHealth nor accelerometry, and 64% of them were published between 2008 and 2013.

Target variables analyzed in the studies

Some studies reported other target variables in addition to PA, even if they were not registered under EMA protocols. Figure 2 shows these complementary target variables, which included, among others, psychological state, body mass index (BMI), or contextual, social and dietary variables.

Figure 2 Other topics or target variables reported in the reviewed studies, different from the behavior of PA.

Psychological state includes: affection, emotions, depression, stress and anxiety. General activity includes: sedentary behavior, time with the smartphone, time reading, eating, active traveling, etc.

Risk of bias assessment

NICE tool was applied to assess the RoB of 92.1% of studies (70 longitudinal prospective studies), the Cochrane tool was applied to 5 experimental studies (6.6%), and ROBINS-I tool was applied to 1 quasi-experimental study. Figure 3 shows a summary of the results of RoB assessment.

Figure 3 Risk of bias (RoB) of the reviewed studies expressed by the number of studies.

(A) Risk of bias of prognostic studies (NICE tool). Representativeness: the study sample represents the population of interest with regard to key characteristics; Loss to follow-up is unrelated to key characteristics; Outcome is adequately measured in study participants; Confounding: important potential confounders are appropriately accounted for limiting potential bias with respect to the prognostic factor of interest. (B) Risk of bias of experimental and quasi-experimental studies (Cochrane tools). Random allocation: random sequence generation; Allocation concealment; Blinding of participants and personnel; Outcome: blinding of outcomes assessment; Missing data: incomplete outcome data.

About half of the longitudinal prospective studies were evaluated as high RoB in three NICE criteria (representativeness, outcome adequately measured and potential confounders appropriately controlled for), and 29% of studies did not provide enough information about loss to follow-up of participants and/or the amount of missing data. Half of the experimental and quasi-experimental studies showed a high RoB regarding the allocation concealment of participants and the amount of missing data in the outcome measure, and half of the studies did not provide enough information about this last RoB.

Discussion

The aim of this study was to systematically review the scientific publications on EMA relating to PA behavior in order to classify the methodologies and to identify the main mHealth technology-based tools and procedures that have been applied during the first 10 years since the emergence of smartphones. We have found enough evidence to use the term mEMA (mobile-based EMA) when mHealth technology is being used together with the EMA methodology for monitoring lifestyle behaviors such as PA, in real time and in natural environments. A total of 76 studies (in 74 articles) on the use of EMA and PA were included for the synthesis. The majority of studies were carried out on healthy adults, lasted around 1 week and applied mHealth technology (39 studies, 51.3%), mainly using smartphones and accelerometers.

EMA, mHealth and mEMA

The term EMA means ecological momentary assessment and was proposed before 2000 (Stone & Shiffman, 1994). We identified 71% of the studies that specifically used “EMA” for assessing PA behaviors. This results indicate that the term EMA is mostly accepted by the scientific community in the field of PA, but that there are some authors who do not use EMA to refer to the same concept. In this sense, the recent proposal of a checklist for reporting EMA studies (CREMAS) in nutrition and PA (Liao et al., 2016) has become very interesting. However, as we have already commented, this checklist becomes very demanding when you want to describe some published studies that, in general terms, comply with the basic features of the EMA methodology. We did not use this checklist because we were interested in identifying studies that have used EMA or similar basic procedures since the emergence of smartphones on the market.

In this sense, we identified half of the reviewed studies (51.3%, 39 studies) using mHealth technology; that is, they mainly used smartphones for collecting EMA data. This coincides with the technological advances of the last decade due to the emergence of smartphones and mHealth technology. Thus, in the 1st years, the studies that did not use any technology to collect data predominated and most were based on paper and pencil (P&P) self-reports. When smartphones were in full swing but were not yet powerful enough to collect and process multiple data in real time, PDAs predominated. In fact, PDAs were also known as handheld PCs and were small computers that made the collection of user self-reports easier than P&P. The improvement was that it allowed EMA records to be more environmentally friendly than the P&P system. In fact, if more objective data were to be collected, accelerometers were also used (6 of 18 studies using PDAs).

Finally, in the most recent years the emergence of mHealth technology in the field of research on PA and health is most noticeable. This has been possible thanks to the rapid evolution and expansion of both smartphones and applications (apps). Current smartphones have much faster processors than PCs 10 years ago and are capable of monitoring lifestyle behaviors using multi-tasking tools. This improvement, along with small and long-lasting batteries has enabled the recordings to have multiple and very precise built-in sensors (Ozdalga, Ozdalga & Ahuja, 2012) and use the smartphone capabilities to be synchronized for assessing or monitoring health or lifestyle (Fiordelli, Diviani & Schulz, 2013; Free et al., 2010).

In other systematic reviews on the application of EMA in PA behavior, only one study used a combination of technology for the EMAs in nutrition and PA among young people (Liao et al., 2016), and only 3 studies with adults combined EMA with mobile phones and accelerometers (Romanzini et al., 2019). In our systematic review, much less restrictive with the concept of EMA, the main built-in sensor used for collecting PA behavior with smartphones was the accelerometer (27 of 39 studies using mHealth). Only 3 studies used GPS (35,89,90) and only 1 study used a cardiac chest band for recording heart rate to collect objective data related to PA (Brannon et al., 2016). Only 3 of these studies used the combination of 2 sensors (Brannon et al., 2016; Seto et al., 2014, 2016). In short, it is clear that the use of mHealth is increasing, but there is still a long way to make the most of the capabilities of this technology for applying EMA to PA behavior.

Thus, based on our systematic review, we have found enough evidence to firmly propose the use of the term mEMA (mobile-based EMA) when mHealth technology is being used together with the EMA methodology, although this term had already been used previously (Spook et al., 2013). More specifically, we agree with defining mEMA as “the use of mobile computing and communication technologies for the EMA of health and lifestyle behaviors”. As an example, some researchers have combined mHealth and EMA methodologies to study alcohol, tobacco and drugs consumption (Serre et al., 2015), depression and anxiety (Schueller, Aguilera & Mohr, 2017), eating disorders and obesity (Engel et al., 2016), or nutrition and PA behaviors (Bruening et al., 2016). Thus, mEMA methodology can facilitate monitoring of healthy lifestyles under both subjective and objective perspectives, using tools such as written diaries or self-reports on a touch screen, messages on social networks, and motion or physiological sensors, all managed through smartphone apps (Shiffman, Stone & Hufford, 2008; Van Os et al., 2017).

Profile of the populations studied with EMA and PA

The selected studies included heterogeneous samples from children to older adults, men and women, healthy and clinical participants. Regarding the age of the participants, the studies were carried out on young-adult samples, including children and adolescents (25%), university students (26%) and adults (45%). In general, this variety of population age is receptive to the use of new technologies like smartphones as well as using this technological advance daily (Bagot et al., 2018; Poscia et al., 2015). This fact could explain the high rate of technology used for EMA assessments in our review (76.3%). Conversely, only 3 studies were found using EMA and PA in older adults (Cabrita et al., 2017; Kanning, Ebner-Priemer & Schlicht, 2015; Kanning & Hansen, 2017). Studies in the elderly are an important challenge for science, due to the increase in life expectancy (World Health Organization, 2018). Thus, we encourage scientists to incorporate older adults and elderly samples in the future EMA projects. It has been pointed out that older people are reluctant to participate in studies that promote electronic forms of data collection (Maher, Rebar & Dunton, 2018). However, technology is becoming an increasingly used strategy in health research (Patrick et al., 2008), because, among other reasons, it may offer a great level of accuracy in PA research using EMA (Myin-Germeys et al., 2018). Thereby, mHealth research has the capability to adapt the advances in technologies to elderly and overcome the initial rejection. In fact, the 3 studies found with older adults used smartphones and accelerometers (Kanning, Ebner-Priemer & Schlicht, 2015; Kanning & Hansen, 2017), and only one of them reported difficulties with the battery management and the size of the response scales on the screen (Cabrita et al., 2017). It is noteworthy that the average age of the participants in the two studies without technological issues was 60.1 ± 7.1, which is not even considered old population in most countries; whereas the average age was 68.7 ± 5.5 years in the only study with few technological difficulties, which could be considered pre-old age (Ouchi et al., 2017). This fact could explain the positive acceptance of technology by the older adult samples analyzed in the 3 selected studies.

As regards gender, we found more studies carried out with women (60%) rather than men. Previous literature has indicated that women could be more adherent than men to participate in research projects and this could be a possible explanation for the gender differences found (Fouad et al., 2014). Interestingly, in general women were less likely to participate in both PA and exercise behaviors (World Health Organization, 2018). Therefore, future EMA and PA studies should take into account possible gender differences and it should continue working along this line, seeking gender equality in scientific studies, as part of the RRI framework (Responsible, Research, Innovation) (Pacifico Silva et al., 2018).

As regards health status, most participants were considered healthy and only 14 studies (18%) included samples with both a physiological or psychological clinical condition. For the latter, the most common were mood disorders. In this sense, current studies show possible relationships between mood disorders and PA. In detail, PA has been presented as a good complemental therapeutic strategy to reduce stress, alleviate depression symptoms and enhance psychological states (Mikkelsen et al., 2017). Thus, EMA methodology could contribute to further explain the relationship between psychological states and PA engagement. In addition, the main topic that accompanied PA in healthy samples was “affect and emotions” (Fig. 2), highlighting the influence of PA on psychological states.

Study designs used in EMA and PA, and methodological aspects

An interesting result from the reviewed studies corresponds to the research designs. This systematic review shows that 92% of the studies followed a longitudinal prospective design, while only 7% of them followed an experimental design, and 1% a quasi-experimental design (Table 2). Hence, increasing the amount of experimental designs could help in determining cause-effect relationships as regards PA and other variables like mental states in future studies. As regards the duration of the EMA assessments, the range was from 1 day (Kanning & Schoebi, 2016) up to 12 months (Burg et al., 2017). The majority of the studies lasted between 1 and 7 days (58%). From this result, it is suggested to increase the duration of the future studies using EMA in order to determine long-term habits. However, a limitation of extended longitudinal studies could be an increase in the number of dropouts (Gustavson et al., 2012). This is known as attrition concern can lead to the subsequent biases of auto-selection and experimental survival (Carmona-Bayonas et al., 2018). In other words, participants who reported all EMA assessments throughout a very long study could have different individual characteristics from those who not complete all the study. It was then checked whether studies showing a high Risk of Bias related to loss to follow-up were those with longer duration, without finding significant results. Therefore, we suggest a duration between 1 and 4 weeks as an optimal balance between habits information and low levels of dropout for future EMA studies.

In relation to the within-day intervals for EMA assessments, 35% of the studies established EMA assessments throughout the whole day, whereas 32% scheduled the assessments according to the availability of the participants; for example, studies with children samples avoided school hours (O’Connor et al., 2017). Interestingly, only 26% of studies were to found directly differentiate between working days and non-working days. The comparison of working days and non-working days enables possible patterns of PA depending on the type of day to be studied (Parrado et al., 2009). Finally, in our systematic review there is a great variety in the range of the number of assessments per day. Between 3 and 7 assessments per day were carried out in 37% of studies, followed by 8 to 12 per day (22%) and 1 to 2 per day (16%). We recommend between 3 and 7 assessments per day. In this context, it has been published that excessive prompts or requests for EMA surveys could increase the number of lost responses, or it could cause participants to respond randomly (Dunton, 2017).

Topics associated with the research in EMA and PA

The main topic (omitting demographic data) that accompanied PA was psychological state (Fig. 2), which is a global variable that includes: affection, emotions, depression, stress or anxiety. This is interesting due to the high rates of mental disorders in the general population worldwide (World Health Organization, 2017) and the positive relation between PA practice and mental health. General activity (as well as BMI) was the second topic studied along with PA. It is related to everyday behaviors like active transport, watching TV or eating. These behaviors are important to understand the lifestyle of the participants and their possible motives and barriers for PA practice (Parrado et al., 2009).

Finally, some stable anthropometric variables like BMI have been assessed in 35 studies (46.1%), although they have not been evaluated by EMA, demonstrating the importance of PA behavior in relation to overweight and obesity (Chin, Kahathuduwa & Binks, 2016).

Limitations

Our systematic review is pioneer in examining mHealth application for EMA studies in the field of PA, though there are some limitations. First, it was not possible to report on the adherence levels to EMA in the participants of most of the studies, because the methodological strategies for reporting EMA data collection were diverse. Similar information has been reported by Liao et al. (2016) in a systematic review on the use of EMA on diet and PA in young population, highlighting the heterogeneity in the EMA data collection methods. We encourage researchers to incorporate results on adherence to EMA interventions as well as the number of dropouts. Although mHealth technologies can help to provide objective EMA recordings, they also have some difficulties. For instance, the high cost of development and maintenance for mHealth apps, the lack of standardization, data management, technical problems, slow connections, and so on are possible problems that should be borne in mind (Kip et al., 2018). In addition, mEMA devices could be expensive if they are only used for research purposes.

Future research directions

mHealth technology could be of great help to apply EMA strategies in developing countries (Hurt et al., 2016), since it reduces the costs compared to a traditional intervention (Chung et al., 2015). A few years ago, specific and expensive sensors like isolated accelerometers were used to investigate PA behaviors.

We encourage researchers to require participants to use their own devices instead of providing specific research instruments to take advantage of the increasing ability of smartphones for synchronously monitoring different objective parameters (Ozdalga, Ozdalga & Ahuja, 2012; Fiordelli, Diviani & Schulz, 2013; Free et al., 2010). This should be the current trend in research on healthy lifestyles like PA behavior. Smartphone apps and sensors (mHealth technology) allow the concept of mEMA to be accepted, especially in relation to PA. It is ideal to use built-in sensors or other sensors easily connected by Bluetooth for monitoring the duration, frequency and intensity of PA. Thus, built-in sensors like accelerometer, GPS, altimeter or gyroscope can provide continuous information about PA running in background while the user performs their daily activities. Or you can also add simultaneous parameters from other external sensors connected via Bluetooth, such as cardiorespiratory information using thoracic bands or other wearables. The tendency should be the automatic recognition of the user activity without interrupting their behavior, based on machine learning algorithms. The ecological information could be completed with voice messages, image captures or brief text selections on the touch screen made in real time. Current mobile devices already have the ability to process all this information, but it will be necessary to persuade the user to carry their smartphone during the whole day.

Conclusions

In our review we have classified the EMA methodologies used for assessing PA behaviors and found that 71% of studies specifically used the term “EMA”. Just over half (51.3%) of studies used mHealth technology, mainly smartphones, for collecting EMA data. An accelerometer was the main built-in sensor used for collecting PA behavior by means of mHealth (69%). The change of trend in the use of tools for EMA in PA coincides with the technological advances of the last decade due to the emergence of smartphones and mHealth technology.

There is enough evidence to use the term mEMA (mobile-based EMA) when mHealth technology is being used together with the EMA methodology for monitoring lifestyle behaviors such as PA, in real time and in natural environments. We define mEMA as the use of mobile computing and communication technologies for the EMA of health and lifestyle behaviors. It is clear that the use of mHealth is increasing, but there is still a long way to make the most of this technology in order to apply EMA to PA behavior. Thus, mEMA methodology can help in the monitoring of healthy lifestyles under both subjective and objective perspectives. The tendency for future research should be the automatic recognition of the user’s PA without interrupting their behavior. From our review, we suggest the use of mEMA methodology with experimental designs, a duration between 1 and 4 weeks as an optimal balance between habits information and low levels of dropout, a number of assessments per day between three and seven, differentiating between working days and non-working days. The ecological information could be completed with synchronized information from other sensors or wearables, all managed through smartphone apps. This methodology could be extended when EMA combined with mHealth are used to evaluate other lifestyle behaviors.

Supplemental Information

Supplemental Information 1 PRISMA checklist.

Click here for additional data file.

Supplemental Information 2 Search strings for all databases in the systematic review.

Click here for additional data file.

Supplemental Information 3 Rationale for conducting this systematic review and contribution to knowledge in light of previously published related reports, including other systematic reviews.

Click here for additional data file.

Additional Information and Declarations

Competing Interests

Author Contributions

Data Availability

The authors declare that they have no competing interests.

Rafael Zapata-Lamana conceived and designed the experiments, performed the experiments, analyzed the data, prepared figures and/or tables, authored or reviewed drafts of the paper, and approved the final draft.

Jaume F. Lalanza performed the experiments, analyzed the data, prepared figures and/or tables, authored or reviewed drafts of the paper, and approved the final draft.

Josep-Maria Losilla conceived and designed the experiments, authored or reviewed drafts of the paper, and approved the final draft.

Eva Parrado analyzed the data, authored or reviewed drafts of the paper, and approved the final draft.

Lluis Capdevila conceived and designed the experiments, performed the experiments, analyzed the data, prepared figures and/or tables, authored or reviewed drafts of the paper, and approved the final draft.

The following information was supplied regarding data availability:

The references for this systematic review were obtained from the following databases and portals: PsycINFO by PsycNET, CINAHL by EBSCOhost, MEDLINE by PubMed, and Core Collection of Web of Science by Web of Science. The data is available in the tables.

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
