# Peer review of "mHealth technology for ecological momentary assessment in physical activity research: a systematic review"

_PeerJ, doi:10.7717/peerj.8848_

## Round 0.1 · original submission · Major Revisions

Both reviewers have provided consistent feedback on the need for better English expression and I agree. Figures 2-4 can also be deleted. Please pay particular attention to the feedback from both reviewers on the need to clarify the aims of the paper in the introduction and abstract.

Reviewer 1 ·

Basic reporting

The present manuscript: mHealth technology for ecological momentary assessment in physical activity research: a systematic review

The manuscript systematically reviews publications on EMA methodologies relating to physical activity behavior in order to identify the main procedures and instruments of applying mHealth technology during the first 10 years since the emergence of smartphones.

Although this is a very interesting paper, the manuscript needs to be revised due to mistakes in language which lead to misunderstandings.
The final conclusion is not clear to me, how the authers come to the new definition of the term mEMA.

Developing the new aspect from the review might be considered as a research aim, next to the systematical review.

The authors should also comment on the aspect if there is a difference in mHealth technology that provides feedback in comparison to technology without feedback function, since this could have an effect on physical activity levels.
Direct feedback might also be an issue in mEMA.


ABSTRACT:
Line 20: The abbreviation EMA should be explained when firstly used.

INTRODUCTION:
Line 60: Please specify “many health disorders”
Line 63: ….evidence can be “contrasted” the term is unclear in this context.
Line 72: Therefore it would be “interesting”…. Please change the term ‘interesting’ and specify the importance
Line 84: ‘when’ should be changed to ‘if’ deleate the word ‘some’ in the sentence
Line 88: delete the word ‘some’
Line 89: Specifically we would like to point out……
Please change the whole sentence, to make the context more clearly
Line 102: delete the word ‘some’
Line 105: the verb is missing in the sentence
Line 110 “massive” the word in over exaggerating

Line 125: “The search comprised the period between 2008 ad 2018…because we were interested in analyzing the use of mHealth…..”
The relationship in this sentence is unclear. Please explain why you selected the time period from 2008 – 2018 specifically?

RESULTS
Line 187/188 it is not clear why you analyzed 76 instead of 74 articles?
Please explain this issue more clearly
Line 195: Why do you explicitly refer to the year 2017 with 19.7% of publications? Please comment.
Please change the ‘comma’ into a ‘point’ in 19,7%

Line 196: When you are referring to publications in North America, please add there the publication rate of the USA. Because you are referring to North America.

Line 202 and Line 204, Line 2017 please delete “old”.
Line 207 Grammar: Studies designs or study design? Singular or plural? Please correct.
Line 217, change ‘between’ to e.g. ‘for a duration of’
Line 222: the word ‘only’ is not understandable? Why ‘only’?

DISCUSSION
Line 287 – Line 303: “Table 1 describes…..” should not be part of the discussion. I would suggest it for the section RESULTS
The whole section is very long for a usually short summary of results at the beginning of a discussion. Please change to a more specific summary or your main results and move the description of results to the section ‘RESULTS’.
Line 307: “76,5%” should be changed to 76.5%
Also the section from line 304 to 312 is a description of results. Please change or comment, why this part should belong in the description of the discussion.
Line 318: the abbreviation PDA is used already in Line 317. Please add the description of PDA to the section where it is mentioned first.

Line 370 … add “year” when referring to the age of the population
Line 378: ‘relevant’ it is unclear what is meant by that? Why is it that the results found are more relevant?
Line 412/412: ‘weekdays’ and ‘non-working days’ this is unclear. Please change to 'working days' or 'non-working days' OR 'week days' and 'week-ends'
Line 427 ‘really’ I would suggest to delete the word, since it exaggerates.

Line 453/454: The sentence is unclear. What is the benefit of using own devices? I recommend to insert a reference for this statement.

CONCLUSION
Lines: 471 – 484 the conclusion is not clear. The authors bring up the new term ‘mEMA‘. The evidence for this new term at the end of the paper is not clearly explained. A new definition of this term is given in the conclusion.
I would recommend to bring it to the introduction of the paper and ask the research question of this term can be used from the reviewed literature?
REFERENCE:
The following references are not correctly cited. Mainly authors are missing: Reference number: 66, 83, 101, 103, 113.

Experimental design

The authors used the PRISMA Statement to design their review. The design is clearly stated.
Four databases were searched from 2008 to February 2018.
A possible research aim towards the new definition "mEMA" should be included in the manuscript.

Validity of the findings

I think the findings are valid.
I do not think that figures 2 – 4 are needed.

Additional comments

The final conclusion is not clear to me:
What exactly is the benefit of mEMA and mHealth and the possible combination of these?
How do you draw these statements from the literature review you have done?

Please further discuss the impact of feedback given by the technologies? Some devices such as accelerometers give immediate feedback on the physical activity level to the user, others do not. Please discuss this issue in terms of EMA and mEMA.

Reviewer 2 ·

Basic reporting

The English used throughout the manuscript was choppy and at times, very difficult to read. The manuscript would benefit from extensive editing for English language to improve clarity.

Experimental design

no comment

Validity of the findings

The Introduction doesn't set up the Conclusions, and could benefit from editing to help the reader better understand the purpose of the manuscript.

Additional comments

The abstract has issues with English word choice, and should be checked over for grammatical errors. This is also generally true for the entire article. The word choice and writing made it difficult at times to discern what the purpose of the article was. The Discussion provides more information on the goals, and I suggest re-writing the Introduction to highlight that this review focused more on methodology and classifying what methods are used in EMA studies, as this is not as clear as it could be.
Introduction
1. Line 65, there are two ‘_ _’ that should be deleted.
2. Line 71, ‘neither’ and ‘nor’ should be ‘either’ and ‘or’.
Methods
1. Why didn’t the authors use index terms during their literature search? It’s possible that they may have missed some articles with their search strategy terms.
2. Line 154, ‘synthesis’ should be ‘synthesize’.
3. Line 178, ‘researches’ should be ‘researchers’.
Results
1. Line 195, ’19,7%’ should be ’19.7%’.
2. Throughout the results, there are errors and inconsistencies with reporting of the percentages. Sometimes a comma is used to delineate a decimal, sometimes a period. Please correct.
3. Line 248, ‘no used’ should be ‘did not use’.
Discussion
1. Lines 280-282: I’m confused about this sentence that the authors found evidence to use the term mEMA. Inventing a new nomenclature didn’t appear to be the main aim of the study in the Introduction, therefore it’s confusing as to why it is relevant to develop this nomenclature.
2. Lines 319-320: This sentence on EMA registrations is difficult to decipher. Please re-word to improve clarity.
3. Line 327, seems to be missing a word in between ‘allowed to’.
4. EMA, mHealth, and mEMA section- this appears to be very repetitive of the Results section, and the authors need to synthesize the results more into overarching conclusions, not simply repeat what can be found in Table 1.
5. Line 375, ‘adherents’ should be ‘adherent’.
Conclusions
1. The font changes from lines 471-483. Please correct.
Figure 1- there is an extra arrow leading from ‘after duplicates removed’.
Figure 2- the font is very small, and it’s fuzzy. Is there a way to correct this? Also, in the text the authors state that the age range for university students overlapped with that of adults. Were the studies that included these age ranges double counted in the figure?
Figures 3-6- the font is very small and fuzzy. Suggest improving figure quality to make it easier to read.

---

## Round 0.2 · Minor Revisions

Thank you for revising your manuscript. Please attend to the minor revisions requested by the reviewer.

Reviewer 2 ·

Basic reporting

Another read-through for minor edits to English language is needed in Discussion.

Experimental design

The methods are well-described and presented in the text as well as in multiple tables/figures. The authors have been responsive to reviewer feedback and improved the clarity of the manuscript objective.

Validity of the findings

Findings appear valid and conclusions and recommendations are supported by findings.

Additional comments

Overall, the authors were very responsive to the suggestions from reviewers. The English language edits have greatly clarified the overall objective and interpretation of findings. This manuscript will be a good contribution to the field.

Results: What was the degree of agreement for paper reviewers?
Line 212- ’13.663’ is too many decimal points. ’13.7’ participants should be sufficient and matches other decimal places reported in this manuscript.
Line 299- I’m unsure why the authors refer to this manuscript as a ‘qualitative synthesis’, and they have used quantitative methods to classify studies.
Another read-through for minor edits to English language is needed in Discussion.
Line 394- ‘prognostic’ should be changed to ‘prospective’
Line 412- the authors state that ’20 studies were found directly differentiate…’. It would be more consistent with other results reported in the Discussion to provide the percentage of studies that differentiate between days. In addition the word ‘to’ is missing between ‘were’ and ‘found’.
Lines 452-467- this reviewer agrees that we should encourage the use of participants own devices. In addition to the rationale for this conclusion mentioned by the authors, the use of a participants own device may be more feasible in older populations because they are familiar with that device, and thereby increase adherence and minimize drop out. The authors may want to consider adding this conclusion to this paragraph of the Discussion.
There are a few cases where the fonts are not consistent throughout the manuscript and references. The references also need to be checked- some include addresses (e.g., reference 44).
Table 1, point 2, definition of PA: ‘any intensity higher than the basal metabolic rate’ also includes sedentary time (defined as basal to 1.5 METs). Therefore I would suggest revising this statement.
Table 2: change ‘,’ to ‘.’ To delineate decimal points to match manuscript text.
Figure 3a appears to have the words on the X axis cut off.

---

## Round 0.3 · accepted · Accept

Thank-you for your minor revisions in response to Reviewer 2.